# CoNiO_2_/Co_3_O_4_ Nanosheets on Boron Doped Diamond for Supercapacitor Electrodes

**DOI:** 10.3390/nano14050474

**Published:** 2024-03-05

**Authors:** Zheng Cui, Tianyi Wang, Ziyi Geng, Linfeng Wan, Yaofeng Liu, Siyu Xu, Nan Gao, Hongdong Li, Min Yang

**Affiliations:** 1State Key Lab of Superhard Materials, College of Physics, Jilin University, Changchun 130012, China; cuizheng21@mails.jlu.edu.cn (Z.C.); w850308519@163.com (T.W.); ziyigeng2021@163.com (Z.G.); wanlf20@mails.jlu.edu.cn (L.W.); lyf21@mails.jlu.edu.cn (Y.L.); xusy21@mails.jlu.edu.cn (S.X.); 2Sichuan Provincial Key Laboratory for Structural Optimization and Application of Functional Molecules, College of Chemistry and Life Science, Chengdu Normal University, Chengdu 611130, China

**Keywords:** BDD substrate, CoNiO_2_/Co_3_O_4_, electrodeposition, supercapacitor electrodes

## Abstract

Developing novel supercapacitor electrodes with high energy density and good cycle stability has aroused great interest. Herein, the vertically aligned CoNiO_2_/Co_3_O_4_ nanosheet arrays anchored on boron doped diamond (BDD) films are designed and fabricated by a simple one-step electrodeposition method. The CoNiO_2_/Co_3_O_4_/BDD electrode possesses a large specific capacitance (214 mF cm^−2^) and a long-term capacitance retention (85.9% after 10,000 cycles), which is attributed to the unique two-dimensional nanosheet architecture, high conductivity of CoNiO_2_/Co_3_O_4_ and the wide potential window of diamond. Nanosheet materials with an ultrathin thickness can decrease the diffusion length of ions, increase the contact area with electrolyte, as well as improve active material utilization, which leads to an enhanced electrochemical performance. Additionally, CoNiO_2_/Co_3_O_4_/BDD is fabricated as the positive electrode with activated carbon as the negative electrode, this assembled asymmetric supercapacitor exhibits an energy density of 7.5 W h kg^−1^ at a power density of 330.5 W kg^−1^ and capacity retention rate of 97.4% after 10,000 cycles in 6 M KOH. This work would provide insights into the design of advanced electrode materials for high-performance supercapacitors.

## 1. Introduction

Due to the increasingly serious problems of air pollution and insufficient resource storage, the development of clean and sustainable energy storage devices is highly pursued. As a kind of electrochemical energy storage device, supercapacitors have advantages of high power density, fast charging and discharging speed, and long cycle life [1,2]. In general, the electrode material is one of determinant factors affecting the performance of supercapacitors, such as capacity and cycling stability in the reaction [3]. According to the energy storage mechanism, the boundary between capacitive behavior (capacitive) and battery behavior (faradaic) is relatively clear: capacitive is physical energy storage, faradaic is chemical energy storage [4]. The difference is whether there is a transition between electrical energy and chemical energy. In other words, whether there is a redox reaction [5]. The electrode materials are classified into electrical double-layer capacitors and pseudocapacitors [5,6]. For electrical double-layer capacitors, carbonaceous materials are usually used as the electrode, which accumulates charge at the electrode-electrolyte interface through reversible ion adsorption under electrostatic interaction [4,7]. On the other hand, pseudocapacitors derive their capacitance from the various available oxidation states for rapid surface or near-surface redox charge transfer [8], transition metal oxides and binary transition metal molybdates are considered as electrode materials for pseudocapacitors because of their high specific capacitance and energy density [9,10]. Improving the energy density of supercapacitors is crucial to meet future practical applications without sacrificing the power density.

Transition metal oxide material Co_3_O_4_ has received widespread attentions as supercapacitor electrodes due to its high theoretical specific capacitance (3560 F g^−1^), low price, environmental friendliness and chemical durability [11,12]. Nevertheless, its poor conductivity limits the transport of electrolyte ions, resulting in a slow electrochemical process of charge storage and delivery [12]. The method of combining a pseudocapacitive material with high conductivity as a framework [13] could effectively increase the utilization of active materials and result in higher capacitance. Due to the redox reaction of nickel and cobalt ions being more abundant than that of a single component of nickel oxide or cobalt oxide (Co^2+/3+^ and Ni^2+/3+^) [14], CoNiO_2_ has higher electrical conductivity and superior electrochemical activity [15]. We suppose that the Co_3_O_4_ and CoNiO_2_ composite structure would have high electrochemical performance.

However, note that both Co_3_O_4_ and CoNiO_2_-based electrodes have low working voltage windows, typically smaller than 0.6 V (Appendix A). The low energy density of supercapacitors is mainly attributed to relatively low voltage window and specific capacity. Based on the formulation: *E* = 1/2*CV*^2^, the energy density (*E*) linearly increases with specific capacitance (*C*) and square of maximum operation voltage (*V*) [16]. Hence, the improvement of maximum operation voltage and specific capacitance is an effective route to increase the specific energy. This strategy has been successfully applied in a hierarchical Co_3_O_4_/MnO_2_ composite material [17]. With the rapid development of chemical vapor deposition methods and the optimization of production processes, the current cost of boron-doped diamond (BDD) is gradually decreasing, so it has a high market value [18]. BDD emerges as a promising electrode material due to high stability in many corrosive media [19,20], low background current and wide potential window in aqueous solution (∼3.2 V) [21,22]. The most noticeable effect related to the sp^3^/sp^2^ ratio on BDD is the variation on the potential window. The sp^2^-bonded carbon playing a modulator role in charge-transfer approaches promoting outer-shell or inner-sphere electron-transfer mechanisms, can effectively capture transport electrons to restrain oxygen evolution at high voltage region, and then construct a supernal potential window [21]. Previously, Jiang et al. [23] used a mesh diamond as an electrode and obtained an energy density of 0.016 W h kg^−1^ and a power density of 9.54 W kg. Yang et al. [24] reported diamond nanoneedle composite graphite as an electrode, an energy density of 0.013 W h kg^−1^ and the power density of 12.79 W kg^−1^ were obtained. Therefore, BDD is chosen as a substrate to enlarge the potential window of a CoNiO_2_/Co_3_O_4_ composite structure, thus the electrochemical performance should be further improved.

Herein, we provide a new strategy for synthesizing CoNiO_2_/Co_3_O_4_ nanosheet arrays on BDD (CoNiO_2_/Co_3_O_4_/BDD) through the simple electrodeposition process. The CoNiO_2_/Co_3_O_4_/BDD sample as a supercapacitor electrode provides high specific capacitance of 214 mF cm^−2^ and long-term cycling stability (85.9% after 10,000 cycles). In addition, an asymmetric supercapacitor is assembled using CoNiO_2_/Co_3_O_4_/BDD and activated carbon as the positive and negative electrodes. This work provides strategic insights for the rational design of supercapacitor electrodes with high areal capacitance and large window voltage.

## 2. Materials and Methods

### 2.1. Synthesis of BDD

BDD samples were synthesized on silicon substrate (Tianjin Jingchen Electronics Company, Tianjin, China) by microwave plasma chemical vapor deposition (MPCVD). Nucleation of the silicon substrate was carried out by suspension containing nanodiamond particles (5–10 nm, Tianjin Qianyu Superhard Technology Co., Ltd., Tianjin, China). First, a small amount of nano-diamond was added on the sandpaper, and ground on the polishing Si surface for half an hour to make even scratches on the surface. Then, the polished Si sheet is put into a mixed solution of acetone and alcohol containing nano-diamond powder for more than half an hour of ultrasound. The nano-diamond impinges on the Si substrate surface through ultrasound to form a micro impact crater, which can reduce the nucleation barrier and increase the nucleation density when growing the diamond. Finally, the samples were ultrasonically cleaned with alcohol and deionized water for 5 min to remove the residual diamond powder on the surface, and then placed on the sample holder to dry. In this way, the nano-diamond was successfully fixed on the silicon substrate. With the ratio of gaseous methane (CH_4_) and hydrogen (H_2_) 5% as the reaction source, under the condition of microwave power of 2200 W and process pressure of 10 kPa, boron source trimethyl borate (C_3_H_9_BO_3_) was introduced using H_2_. The substrate was heated to about 850 °C using an induction heater, and the substrate temperature was measured using a thermocouple. After deposition for 12 h, the thickness of BDD film was about 20 μm. The mass of hydrogen consumed to obtain a BDD film with a thickness of 20 μm was 25.63 g. According to the test of Hall effector, the conductivity of BDD was 113.63 S cm^−1^.

### 2.2. Synthesis of CoNiO_2_/Co_3_O_4_/BDD

CoNiO_2_/Co_3_O_4_ arrays were conducted on BDD by using a CHI 760E model Electrochemical Workstation in a standard three-electrode system. Ni(NO_3_)_2_·6H_2_O (0.8 mmol), Co(NO_3_)_2_·6H_2_O (0.8 mmol) and NH_4_Cl (8 mmol) were mixed in 80 mL deionized water and transferred into 100 mL electrolytic cell. With BDD as the working electrode, Pt sheet as the counter electrode and Ag/AgCl electrode as the reference electrode, a three-electrode system of potentiostatic electrodeposition was constructed. The effect of NiCo precursor on 1000 s, 2500 s, and 5000 s reaction time was investigated at constant voltage of −1.0 V. Finally, the CoNiO_2_/Co_3_O_4_/BDD hybrid structure was obtained by annealing for 2 h at a heating rate of 5 °C min^−1^ in an argon atmosphere at 300 °C.

### 2.3. Materials Characterization

The morphology of the samples was investigated with a scanning electron microscope (SEM, FEI Magellan 400, Hillsboro, OR, USA). Transmission electron microscopy (TEM, JEOL JEM-2100FS, Tokyo, Japan) and energy-dispersive X-ray spectroscopy (EDS) were used to study the microstructures. X-ray photoelectron spectroscopy (XPS) analysis was performed with the VGESCALAB MK II system (Uppsala, Sweden) using a monochromatic AlKa (1486.6 eV) X-ray source under ultra-high vacuum (background pressure: 4.4 × 10^−9^ mBar). The crystalline phases of the samples were determined using X-ray diffractometer (XRD, SmartLab, Rigaku, Tokyo, Japan). The average thickness of samples was estimated by atomic force microscopy (AFM, Cypder ES, Oxford, UK). Cyclic voltammetry (CV), galvanostatic charge-discharge (GCD), and electrochemical impedance spectroscopy (EIS) of the prepared CoNiO_2_/Co_3_O_4_/BDD electrode were performed on CHI760E electrochemical workstation (Chenhua, Shanghai, China).

### 2.4. Negative Electrode Preparation and Assembly of Asymmetrical Supercapacitor

The negative electrode was prepared by mixing activated carbon (specific surface area: 1800 m^2^ g^−1^, granularity: 5–8 μm) purchased from Aladdin Industrial Corporation (Shanghai, China), acetylene black (conductive additive), and polyvinylidene fluoride (PVDF) binder in ethyl alcohol solvent with a weight ratio of 8:1:1. The slurry was then coated onto a nickel foam current collector (Suzhou Kesheng and metal Materials Co., Ltd., Suzhou, China) (1 cm × 1 cm) and dried for 24 h under a vacuum at 80 °C. The two-electrode structure of an asymmetric supercapacitor assembled with CoNiO_2_/Co_3_O_4_/BDD electrode, activated carbon as negative electrode and separator (NKK-MPF30AC-100, Tokyo, Japan) was tested in 6 M KOH solution (Aladdin Industrial Corporation, Pico Rivera, CA, USA). Appendix A shows the supercapacitor structure diagram. The mass ratio of two electrodes was balanced by the relationship:m_+_/m_−_ = (C_−_ × ΔE_−_)/(C_+_ × ΔE_+_)(1)
where m (g) was the mass of the electrode materials (anode or cathode), C (F/g) was the specific capacitance, and ΔE was the potential window.

## 3. Results

The synthesized BDD film is composed of dense grains with uniform grain size, and the grains are free of cracks and holes (Figure 1a). Figure 1b–d shows the SEM images of annealed CoNiO_2_/Co_3_O_4_/BDD composites with deposition time of 1000 s, 2500 s and 5000 s, respectively. The thickness of CoNiO_2_/Co_3_O_4_/BDD nanosheet can be accurately measured by AFM. The vertical coordinate represents the roughness of the sample, and the horizontal coordinate represents the thickness of the sample. After 1000 s of deposition, a crisscrossing skeleton structure is found (Figure 1b). Layer thickness of roughly is 10 nm by AFM analysis (Appendix A). Figure 1c shows that the vertically arranged CoNiO_2_/Co_3_O_4_/BDD nanosheets grow dense, when the reaction deposition time is 2500 s. As shown in Appendix A, the thickness of the nanosheet structure is about 20 nm. The heterostructure consisting of two vertically aligned interconnected hierarchical nanosheets not only provide a large surface area and an efficient diffusion pathway for fast electron/ion transport, but also generate abundant electron-altering heterointerface, resulting in a synergistic effect between the two components [25]. When the reaction proceeds further to 5000 s, the morphology of CoNiO_2/_Co_3_O_4_ is obviously agglomerated, exhibiting a pronounced massive structure (Figure 1d).

Since thicker and denser structures are beneficial to improve structural stability and ion transport efficiency during the reaction process [26,27], thus we suppose that the CoNiO_2_/Co_3_O_4_/BDD structure with deposition time of 2500 s has the best electrochemical performance. To confirm this conclusion, CV, GCD and EIS results of electrodes with reaction times of 1000 s, 2500 s and 5000 s in Appendix A demonstrate that the sample with reaction time of 2500 s has the largest CV area, longest discharge time and lowest diffusion resistance. Thus, the CoNiO_2_/Co_3_O_4_/BDD sample with deposition time of 2500 s is used in the following.

To further characterize the structure of the synthesized sample, Figure 2 shows TEM image, high-resolution TEM image (HRTEM) and corresponding EDS mapping analysis results of CoNiO_2_/Co_3_O_4_ powder scraped from BDD substrate. CoNiO_2_/Co_3_O_4_ displays a typical two-dimensional nanosheet morphology with a smooth surface, and the measured layer spacing is 94 nm (Figure 2a). Proper layer spacing is conducive to ion embedding [8]. In HRTEM image (Figure 2b), the interplanar spacings of 0.122 nm and 0.202 nm are indexed to the CoNiO_2_ (222) and Co_3_O_4_ (400) planes, respectively, indicating the coexistence of CoNiO_2_ and Co_3_O_4_ in the composite. Moreover, the EDS elemental mapping images in Figure 2c verify the presence of O, Ni and Co elements in the samples, further proving the successful synthesis of CoNiO_2_ and Co_3_O_4_ heterostructures. Atomic ratio from EDS data is listed in Appendix A.

Figure 3a displays the XRD pattern of CoNiO_2_/Co_3_O_4_ powder scraped from BDD substrate. The characteristic peaks are located at 36.8°, 42.8°, 61.7° and 73.9°, which are indexed to (111), (200), (220) and (311) planes of CoNiO_2_ (JCPDS No. 10-0188). Furthermore, the diffraction peaks at 19.0°, 31.2°, 36.8°, 38.5°, 44.8°, 49.1°, 55.6°, 59.3°, 65.2°, 68.6°, 69.7° and 74.1° correspond to (111), (220), (311), (222), (400), (331), (422), (511), (440), (531), (442) and (620) planes of Co_3_O_4_ (JCPDS No. 43-1003). The more detailed elemental composition and the oxidation state of the prepared CoNiO_2_/Co_3_O_4_ are further characterized by XPS measurements and the corresponding results are presented in Figure 3b–d. Binding energy is calibrated by fixing the saturated hydrocarbon component of the C1s peak at 284.8 eV [28]. Gaussian function fitting is performed for all peaks [29]. The Co 2p^3/2^ and Co 2p^1/2^ spectra can be fitted to two spin-orbit doublets of Co^2+^ and Co^3+^. The Co^2+^ peaks are centered at binding energies of 798.7 and 782.8 eV, and the other peaks at 781.0 and 796.8 eV are attributed from Co^3+^ [30]. These results confirm that the Co species have Co^2+^/Co^3+^ in the hybrid electrodes. In Figure 3c, the Ni 2p spectra display Ni 2p^3/2^ and Ni 2p^1/2^ with two shakeup satellites. In addition, the peaks of Ni 2p^3/2^ and Ni 2p^1/2^ can be broadened by several peaks, whose binding energies center at 855.1, 856.2, 872.4 and 873.6 eV [31], indicating the coexistence of Ni^2+^ and Ni^3+^ in hybrid electrodes. The O 1s spectrum in Figure 3d has two obvious peaks at 532.2 eV and 531.3 eV, which represents metal-oxygen bonds and surface physically adsorbed water, respectively [32]. Thus, the sample surface is mainly composed of Ni^2+^, Ni^3+^, Co^3+^, Co^2+^ and O^2−^ ions, which provides abundant faraday reaction sites and is beneficial to achieve excellent electrochemical performance [32,33]. In addition, we provide XPS of BDD films in Appendix A. BDD has a small amount of sp^2^ carbon, which is conducive to improve the conductivity of the electrode.

In order to highlight the advantages of CoNiO_2_/Co_3_O_4_/BDD electrode, CV, GCD and EIS curves of pristine BDD and CoNiO_2_/Co_3_O_4_/BDD electrodes are presented in Figure 4a–c. The CV curves are performed at a scan rate of 5 mV s^−1^ in 1 M Na_2_SO_4_ electrolyte in Figure 4a. CoNiO_2_/Co_3_O_4_/BDD show a pair of relative symmetrical redox peaks in the voltage window range of 0–1.2 V, indicating faradaic reactions occurred during CV process and good reversibility [34]. It is attributed to the reversible redox reaction of Co and Ni. These associated redox reactions are the transitions of different chemical valences of Co and Ni (Co^2+/3+^ and Ni^2+/3+^) and correspond to the CV peaks of CoNiO_2_/Co_3_O_4_/BDD, which would further increase the capacitance. BDD shows a rectangular CV curve, indicating the double layer capacitance process of CV [6]. It is also shown that the enclosed area of CV for CoNiO_2_/Co_3_O_4_/BDD is larger than that of BDD, suggesting the higher electrochemical capacity of CoNiO_2_/Co_3_O_4_/BDD. The capacities of CoNiO_2_/Co_3_O_4_/BDD and BDD are evaluated by GCD analysis at 1 mA cm^−2^ in Figure 4b. The GCD of BDD has an almost symmetrical linear shape, indicating the typical behavior of an ideal electrical double layer capacitor [35]. The GCD of CoNiO_2_/Co_3_O_4_/BDD has two distinct potential plateaus, confirming that faradaic reduction reactions occur during GCD process [36]. The GCD results match well with the CV results. In addition, the discharge time of CoNiO_2_/Co_3_O_4_/BDD (257 s) is much longer than that of BDD (0.7 s), denoting the higher specific capacity of CoNiO_2_/Co_3_O_4_/BDD.

EIS measurements are performed at open circuit potential in the frequency range of 0.01–100 kHz using a 6 M KOH electrolyte in Figure 4c. The CoNiO_2_/Co_3_O_4_/BDD electrode shows linear in the Nyquist plot, indicating that the active material is completely dispersed in the electrolyte, which further confirms that the CoNiO_2_/Co_3_O_4_/BDD electrode has ideal rate capability [14]. In addition, improving the wettability is an effective method to improve their capacitive performance [37]. As shown in Appendix A, CoNiO_2_/Co_3_O_4_/BDD structure (42°) has a smaller hydrophilic angle than pristine BDD (94°), due to its unique layer structure and better wetting ability in the electrolyte.

Figure 4d shows the CV curves of CoNiO_2_/Co_3_O_4_/BDD at various scan rates. The response current increases linearly with the increase of scan rate. The lower the scanning rate, the better the CV symmetry, the closer to the double layer mechanism, and the better the reversibility [34]. In particular, CoNiO_2_/Co_3_O_4_/BDD extends the voltage window to 1.2 V, which is larger than that of CoNiO_2_ and Co_3_O_4_ based pseudocapacitive electrodes (Appendix A), due to the high potential window of diamond substrate [15,25,38]. The GCD curves in Figure 4e exhibit a symmetrical shape over a wide current density range of 1 to 10 mA cm^−2^, revealing high coulombic efficiency and electrochemical capacitive characteristics due to the highly reversible redox reactions of CoNiO_2_/Co_3_O_4_/BDD electrode during charge-discharge process. The area specific capacitance *C*_s_ (μF cm^−2^) is calculated from the GCD curves according to equation [39]:(2)$$C_{s}=\frac{i\times \Delta t}{s\times \Delta V}$$
where *i* (A) is the current rate of charge and discharge, Δ*t* (s) is the discharge time, *s* (cm^2^) is the effective area of the electrode, and Δ*V* (V) is the voltage window. Then the specific capacitance value of CoNiO_2_/Co_3_O_4_/BDD is largest (214 mF cm^−2^) at the current density of 1 mA cm^−2^ (Figure 4f), being larger than that for Co_3_O_4_/BDD (124 mF cm^−2^) in Appendix A. Notably, the specific capacitance of CoNiO_2_/Co_3_O_4_/BDD exceeds most diamond-based supercapacitors (Appendix A) [18]. In addition, the electrode achieves superior cycling stability, a capacity retention of 85.9% and coulombic efficiency of 99.3% are obtained after 10,000 cycles (Appendix A). The percentages of the capacitive and diffusion contributions can be further quantified by the following Equation (3):(3)$$\;i\left(v\right)=k_{1}v+k_{2}v^{1/2}$$
where *k*_1_ and *k*_2_ are arbitrary constants, and *k*_1_*v* and *k*_2_*v*^1/2^ correspond to capacitive processes and diffusion-controlled effects, respectively. At a scan rate of 1.0 mV s^−1^, the capacitive contribution is 48%, and the diffusion-controlled process accounts for 52% (Appendix A). As shown in Appendix A, with the increase in the scan rate, the capacitive contribution is even higher. This suggests that the capacitive contribution plays a dominant role in the total capacity, and a faraday redox reaction occurs mainly on the surfaces of CoNiO_2_/Co_3_O_4_/BDD nanostructures.

Finally, to verify the energy storage performance of CoNiO_2_/Co_3_O_4_/BDD electrode in practical application, an asymmetric supercapacitor device is assembled using CoNiO_2_/Co_3_O_4_/BDD as the positive electrode and activated carbon as the negative electrode. The CV curves in Figure 5a show the mixed capacity of the electric double-layer capacitor and pseudocapacitance. When the scan rate increases, the area of CV curve enlarges, and the peak shape has good symmetry (Figure 5b), indicating a high coulombic efficiency and good reversibility for the fast charge/discharge process. As shown in Figure 5c, the specific capacitance is largest (79.1 mF cm^−2^ at 2 A·cm^−1^), and the corresponding mass specific capacitance is presented in Appendix A. Furthermore, the almost symmetrical GCD curves indicate high coulombic efficiency and electrochemical reversibility. As shown in Figure 5d, after 10,000 cycles, the device maintains 97.4% specific capacity and coulombic efficiency of 90.8%. In the process of cycling, electrolyte ions have deep adsorption and intercalation in the layer, which changes the structure and leads to the degradation of the electrode structure. At the same time, impurities in the electrode or electrolyte may cause side reactions and affect the stability of the cycle [40]. The energy density *E* (Wh kg^−1^) and power density *P* (kW kg^−1^) of the device are calculated by Equations (4) and (5), respectively [11]
(4)$$E=\frac{1}{2}C_{g}{∆V}^{2}$$
(5)$$\;P=\frac{E}{\Delta t}$$
where *C*_g_ (F g^−1^) is the specific capacitance. Ragone for energy storage devices is intuitive and meaningful [41]. Appendix A also summarizes the energy and power densities of other diamond electrode materials available in the literature. The CoNiO_2_/Co_3_O_4_/BDD device can deliver a maximum energy density of 7.5 W h kg^−1^ at 330.5 W kg^−1^, and 1.2 W h kg^−1^ at 1098.1 W kg^−1^ (Appendix A). This value is higher than graphite@NDD//graphite@NDD (NDD, nano-needles diamond) [23], BDD//BDD [35] and porous BDD//porous BDD [42]. A red light-emitting diode is lit directly by only one asymmetrical supercapacitor without any other power assistance, indicating that this electrode material has excellent application prospects.

## 4. Conclusions

In summary, CoNiO_2_/Co_3_O_4_ nanosheet arrays are synthesized on BDD with the enhanced electrochemical performance by one-step electrodeposition strategy, and used for supercapacitor electrodes. The surface morphology of CoNiO_2_/Co_3_O_4_ can be controlled by the deposition time, and an optimum deposition time is 2500 s. The CoNiO_2_/Co_3_O_4_/BDD electrode shows an excellent capacitance value of 214 mF cm^−2^ along with a voltage window of 1.2 V. In addition, the electrode achieves superior cycling performance stability (a capacity retention of 85.9% after 10,000 cycles). Finally, the assembled symmetric supercapacitors device with CoNiO_2_/Co_3_O_4/_BDD as the positive electrode has an energy density of 7.5 W h kg^−1^, when the power density is 330.5 W kg^−1^, and the capacitance maintains 97.4% of the initial value after 10,000 cycles. The improvement in electrochemical performance is attributed to CoNiO_2_/Co_3_O_4_ with unique two-dimensional nanosheet structure, improved electrical conductivity and BDD with a wide voltage window.

## Figures and Tables

**Figure 1 nanomaterials-14-00474-f001:**
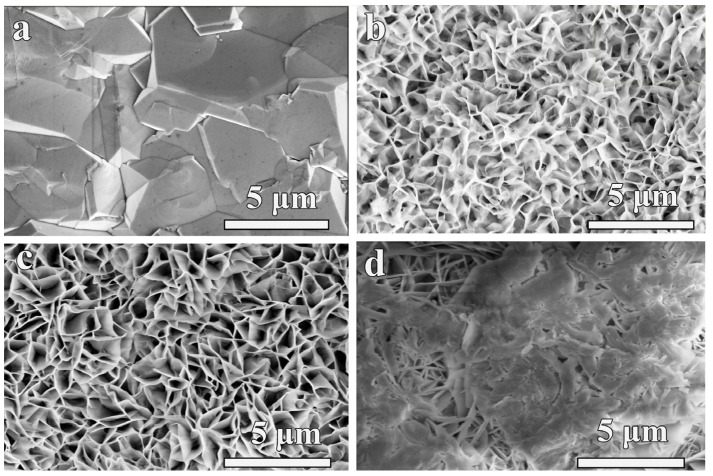
SEM images of (**a**) pristine BDD film, CoNiO_2_/Co_3_O_4_/BDD at deposition time of (**b**) 1000 s, (**c**) 2500 s, and (**d**) 5000 s.

**Figure 2 nanomaterials-14-00474-f002:**
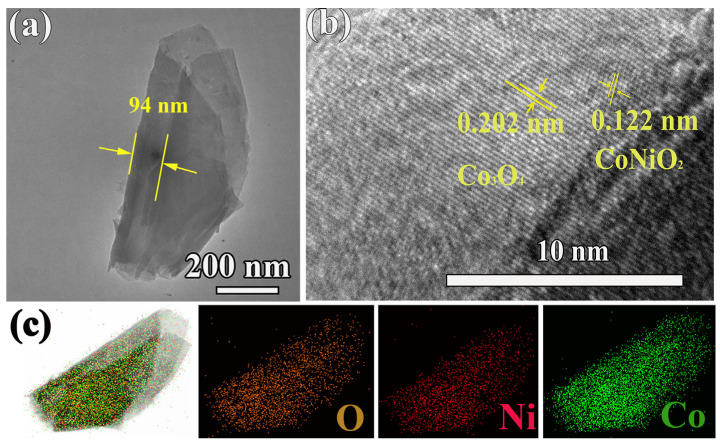
(**a**) TEM image, (**b**) HRTEM image, and (**c**) TEM-EDS elemental mapping images of CoNiO_2_/Co_3_O_4_ powder scraped from BDD substrate.

**Figure 3 nanomaterials-14-00474-f003:**
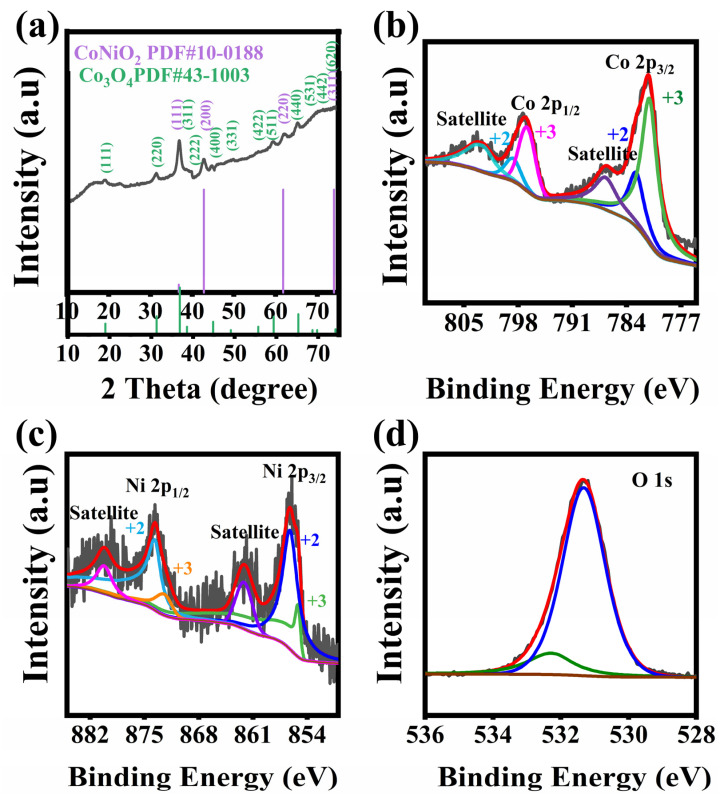
(**a**) XRD pattern and XPS spectra of (**b**) Ni 2p, (**c**) Co 2p, (**d**) O 1s for CoNiO_2_/Co_3_O_4_ powder scraped from the BDD substrate.

**Figure 4 nanomaterials-14-00474-f004:**
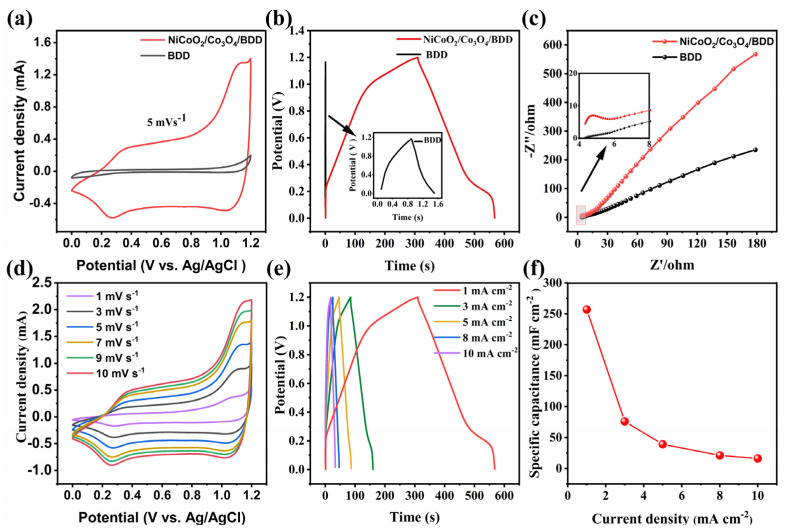
(**a**) CV curves at a scanning rate of 50 mV s^−1^, (**b**) GCD curves at a scanning rate of 1 mA cm^−2^, and (**c**) Nyquist plots of BDD and CoNiO_2_/Co_3_O_4_/BDD electrodes and equivalent circuit. (**d**) CV, (**e**) GCD, (**f**) specific capacitances of CoNiO_2_/Co_3_O_4_/BDD electrode.

**Figure 5 nanomaterials-14-00474-f005:**
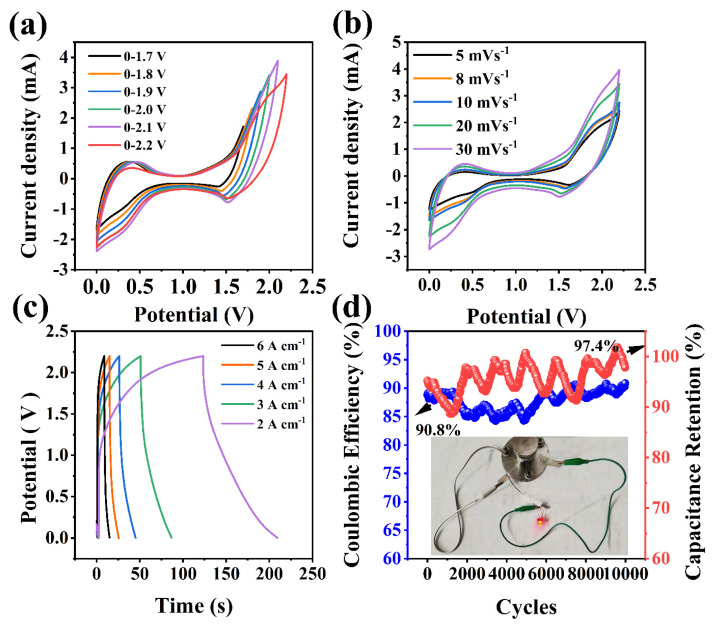
(**a**) CV curves at various voltage ranges (20 mV s^−1^), (**b**) CV curves at various scan rates, (**c**) GCD curves at different current densities, (**d**) cycling stability of CoNiO_2_/Co_3_O_4_/BDD at 5 mA cm^−2^ for asymmetric supercapacitor device using CoNiO_2_/Co_3_O_4_/BDD as the positive electrode and AC as the negative electrode. Inset is photograph of a red light-emitting diode lit for several minutes with a cell device.

## Data Availability

The data presented in this study are available on request from the corresponding author.

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
