# Peer review of "CoNiO2/Co3O4 Nanosheets on Boron Doped Diamond for Supercapacitor Electrodes"

_nanomaterials, 2024, doi:10.3390/nano14050474_

Round 1
Reviewer 1 Report
Comments and Suggestions for Authors
Energy storage systems such as rechargeable batteries and hybrid supercapacitors have received great attention in recent decades, especially in industries like hybrid electric vehicles and smart portable electronics. Supercapacitors store energy through fast ion adsorption at the electrode/electrolyte interfaces or reversible faradaic reactions at the surface layer of electroactive traditional metal oxides and sustain excellent cycling stability. Several different classes of diamond-based materials have been developed for supercapacitor applications. In the submitted work, boron-doped diamond electrode substrates are discussed with improved mechanical properties that can provide better adhesion and stability, where nanosheet arrays are anchored.
The reported work could be novel but the work in its present form is not publishable. It requires major revision.
· In the introduction, lines 32- 33 and 35 -66 mechanisms relating to EDLC and pseudocapacitors must be differentiated.
· Line 38, not only TMO but also binary transition metal molybdates reported recently by Manickam Minakshi et al have also shown to exhibit SC and Energy density.
· Line 46, what is meant by the phrase “high conductivity as a skeleton” does it mean framework?
· The oxidation state/valence charges of Co and Ni must be mentioned in the material CoNiO2.
· Line 50, both the materials have Co in it, is this not overlapping the energy level?
· Is BDD cost-effective?
· Line 92, why Ar atmosphere?
· Provide some details on activated carbon.
· Figure 1d, why the surface of the EM image for 5000s have patched areas?
· Is the synthesized material porous?
· Why two different electrolytes have been used (Line 180; and Line 197) Na2SO4 and KOH? This is very strange. Neutral versus highly basic having 6M.
· The key papers published recently on the supercapacitors discussing their redox behavior and symmetrical peaks such as doi.org/10.1016/j.progsolidstchem.2023.100390; and doi.org/10.1016/j.progsolidstchem.2023.100390 must be cited and explained.
· The reversibility of the CV curves in Figures 4a and 4d must be explained.
· Please provide the SC values for both the 2-electrode and 3-electrode systems.
· The shape of the CV curve in Figure 5a is very weird, which is not a typical CV curve for a full cell (device). Justify.
· Why both the red and blue curves in Figure 5d are like a sinusoidal wave rather than a flat curve?
· The performance metrics of the asymmetric capacitor must be compared with the literature value.
Comments on the Quality of English LanguageThe language must be improved and a few of the lines require clarity.
Author Response
Answers to Reviewers
Reviewer #1
The Energy storage systems such as rechargeable batteries and hybrid supercapacitors have received great attention in recent decades, especially in industries like hybrid electric vehicles and smart portable electronics. Supercapacitors store energy through fast ion adsorption at the electrode/electrolyte interfaces or reversible faradaic reactions at the surface layer of electroactive traditional metal oxides and sustain excellent cycling stability. Several different classes of diamond-based materials have been developed for supercapacitor applications. In the submitted work, boron-doped diamond electrode substrates are discussed with improved mechanical properties that can provide better adhesion and stability, where nanosheet arrays are anchored.
The reported work could be novel but the work in its present form is not publishable. It requires major revision.
The authors thank Referee for offering valuable suggestions and comments to improve the scientific level of the manuscript. We have carefully considered these comments and answered the comments in the following.
Q1: In the introduction, lines 32-33 and 35-66 mechanisms relating to EDLC and pseudocapacitors must be differentiated.
Answer: Thanks for reviewer’s kind comments. According reviewer’s suggestion, we have added the distinction between pseudocapacitors and double-layer capacitors in lines 33-38 of page 1.
Q2: Line 38, not only TMO but also binary transition metal molybdates reported recently by Manickam Minakshi et al have also shown to exhibit SC and Energy density.
Answer: Thanks for reviewer’s kind comments. We have added the discussion of binary transition metal molybdates reported recently by Minakshi in lines 43-45 of page 2.
Q3: Line 46, what is meant by the phrase “high conductivity as a skeleton” does it mean framework?
Answer: Yes, we have revised it to “high conductivity as a framework” in lines 52-53 of page 2.
Q4: The oxidation state/valence charges of Co and Ni must be mentioned in the material CoNiO2
Answer: Thanks for reviewer’s kind comments. According reviewer’s suggestion, we have added Co and Ni oxidation state/valence in the material CoNiO2. The relevant description is in lines 55-56 of page 2.
Q5: Line 50, both the materials have Co in it, is this not overlapping the energy level?
Answer: Actually, the two materials are only compounded together, the heterostructure is formed and the synergistic effect occurs, and the overlap of energy levels is not produced.
Q6: Is BDD cost-effective?
Answer: Yes, BDD is cost-effective. BDD is considered as an excellent electrode material due to its wide potential window, excellent chemical stability and repeatability. With the rapid development of chemical vapor deposition methods and the optimization of production processes, the current cost of BDD is gradually decreasing, so it has a high market value. The relevant description is in lines 66-68 of page 2.
Q7: Line 92, why Ar atmosphere?
Answer: Because argon is recognized as an inert gas, its chemical properties are very stable, in the process of heating, it is not easy to react with the sample, thus, the sample is more crystalline.
Q8: Provide some details on activated carbon
Answer: Thanks for the reviewer’s kind suggestion. Details of activated carbon have been provided in lines 137-139 of page 3. Activated carbon (specific surface area: 1800 m2 g−1, granularity: 5-8 μm) is purchased from Aladdin Industrial Corporation (Shanghai, China).
Q9: Figure 1d, why the surface of the SEM image for 5000s have patched areas?
Answer: Because the electrochemical deposition time is too long, the layer structure of the sample would agglomerate at 5000 s, the morphology of patches forms on the surface.
Q10: Is the synthesized material porous?
Answer: Yes, the synthesized material can be considered as a porous material. The synthesized materials have a two-dimensional structure formed by many polygonal holes on the plane, which is similar to the hexagonal structure of a beehive.
Q11: Why two different electrolytes have been used (Line 180; and Line 197) Na2SO4 and KOH? This is very strange. Neutral versus highly basic having 6M.
Answer: Because in testing device performance, it is also necessary to take into account the synergistic effect of negative and positive electrode materials. KOH can be completely dissociated in aqueous solution. K+ and OH- ions in it can provide conductance function, making a good conductance channel formed in the electrolyte. At the same time, the potassium hydroxide molecule is relatively small and has good fluidity, which can quickly form an electric double layer between the poles of the capacitor.
Q12: The key papers published recently on the supercapacitors discussing their redox behavior and symmetrical peaks such as doi.org/10.1016/j.progsolidstchem.2023.100390; and doi.org/10.1016/j.progsolidstchem.2023.100390 must be cited and explained.
Answer: Thanks for the reviewer’s kind suggestion. According to reviewer’s suggestion, we have discussed the relationship between redox behavior and symmetry peaks according to the reference provided, and the reference is cited as No. 34. The relevant description is in lines 221-224 of page 6.
Q13: The reversibility of the CV curves in Figures 4a and 4d must be explained.
Answer: Thanks for the reviewer’s kind suggestion. According to reviewer’s suggestion, the lower the scanning rate, the better the CV symmetry, the closer to the double layer mechanism, and the better the reversibility. The relevant description is in lines 246-249 of page 7.
Q14: Please provide the SC values for both the 2-electrode and 3-electrode systems.
Answer: Thanks for the reviewer’s kind suggestion. According to reviewer’s suggestion, SC values for 2-electrode and 3-electrode are added in lines 259-260 and 286-287, and the values are 214 mF cm-2 at 1 A·cm−1 and 79.1 mF cm-2 at 2 A·cm−1, respectively.
Q15: The shape of the CV curve in Figure 5a is very weird, which is not a typical CV curve for a full cell (device). Justify
Answer: Thanks for the reviewer’s kind suggestion. We have examined the CV curve in Figure 5a. The shape of the CV curve is determined by the assembly of asymmetric devices with activated carbon as the negative electrode and CoNiO2/Co3O4/BDD as the positive electrode. As can be seen from the GCD curve in Figure 5c, it is close to polarization at 2.2 V during the charging process, indicating that the device is full.
Q16: Why both the red and blue curves in Figure 5d are like a sinusoidal wave rather than a flat curve?
Answer: Because in the process of cycling, electrolyte ions have deep adsorption and intercalation in the layer, which changes the structure and leads to the degradation of the electrode structure. At the same time, impurities in the electrode or electrolyte may cause side reactions and affect the stability of the cycle. The relevant description is in lines 291-294 of page 8.
Q17: The performance metrics of the asymmetric capacitor must be compared with the literature value
Answer: Thanks for reviewer’s kind comments. According reviewer’s suggestion, we have compared the data with other literature in Table S3. The relevant description is in lines 299-305 of page 8.

Reviewer 2 Report
Comments and Suggestions for Authors
See attached document

Comments on the Quality of English LanguageMinor editing required
Author Response
Reviewer #2
The authors describe the fabrication and application of CoNiO2/Co2O3/boron doped diamond films as supercapacitor electrode materials. They reported a specific capacitance of 214 mFcm-1 with an energy density of 7.53 Wh Kg-1. However, the manuscript is not suitable for publication in its current state. I recommend a major revision of the manuscript because of the following reasons:
The authors thank Reviewer for offering valuable suggestions and comments to improve the scientific level of manuscript. We have carefully considered these comments and suggestions and answered the questions accordingly.
Q1: The abstract needs to be reviewed. The writers should describe the features of the nanomaterials that allow them to exhibit superior properties. Also, the electrolyte used should be noted.
Answer: Thanks for reviewer’s kind comments. According reviewer’s suggestion, the abstract have been revised. Nanosheet materials with an ultrathin thickness can decrease the diffusion length of ions, increase the contact area with electrolyte as well as improve active material utilization, which leads to an enhanced electrochemical performance. And the electrolyte is noted. The relevant description is in lines 16-21 page of 1.
Q2: In the introduction section, the authors have mentioned that ..[…a new strategy for synthesizing CoNiO2/Co2O3 nanosheet arrays on BDD (CoNiO2/Co3O4/BDD)…], Was Co2O3 synthesized or was it Co3O4? This needs to be clarified.
Answer: Thanks for reviewer’s kind comments. The synthesized sample is Co3O4, and we have modified it to Co3O4 in lines 82-83 page of 2.
Q3: The authors are interchanging CoNiO2/Co3O4/BDD and CoNiO2/Co2O3/BDD, Yet, the XRD data given shows the formation of Co3O4.
Answer: Thanks for reviewer’s kind comments. Sorry for our mistake, we have carefully checked the whole paper carefully and modified it to Co3O4.
Q4: The XPS data in Figure 3b-3d is too crowded and the different colours are not well presented. This needs to be redone.
Answer: Thanks for reviewer’s kind comments. According reviewer’s suggestion, we have redrawn the XPS in Figure 3.
Q5: Figure S3 illustrates the unstable cycling behavior of CoNiO2/Co3O4/BDD. Why was this the case?
Answer: Thanks for reviewer’s kind comments. Because in the process of cycling, electrolyte ions have deep adsorption and intercalation in the layer, which changes the structure and leads to the degradation of the electrode structure. At the same time, impurities in the electrode or electrolyte may cause side reactions and affect the stability of the cycle.
Q6: In Figure S5, the GCD curves for CoNiO2/Co3O4/BDD demonstrate a considerable IR reduction. This should be explained.
Answer: Thanks for reviewer’s kind comments. Because diamond has a relatively large resistance, there will be a significant IR reduction.
Q7: For comparative purposes, the electrochemical performances of Co3O4/BDD or CoNiO2/BDD should also be assessed
Answer: Thanks for reviewer’s kind comments. According reviewer’s suggestion, we have performed the GCD curves of Co3O4/BDD electrode in Figure S6, the area specific capacitance is 124 mF cm-2. The relevant description is in lines 259-261 of page 7.
Q8: The authors should compare the electrochemical performance of the electrode materials to Co3O4 and similar materials in the literature.
Answer: Thanks for reviewer’s kind comments. According reviewer’s suggestion, the electrochemical performance of the electrode materials to Co3O4 is shown in Table S1. Cobalt-based compounds have a high capacity due to their high theoretical specific capacitance. However, due to the low potential window, the combination of diamond with a wider potential window will provide a good idea for improving the energy density. The related illustration is in lines 59-60 of page 2.

Reviewer 3 Report
Comments and Suggestions for Authors
Review Manuscript ID nanomaterials-2898450
A rather complex design of an asymmetric supercapacitor with rather modest values of specific energy and specific power is described. However, the results obtained appear to be reliable. Once the following issues have been addressed, the manuscript can be published.
1. In the introduction, it is necessary to explicitly indicate whether BDD films were previously used to create supercapacitors, and if they were, then indicate these works and the specific energy and power values ​​obtained in them. This is necessary to understand both the degree of novelty and the level of results obtained in the work being reviewed.
2. For an objective assessment of the work done, in section “2.1 Synthesis of boron doped diamond (BDD)” it is necessary to give the mass of hydrogen that must be consumed to obtain BDD film with a thickness of 20 μm.
3. It is also advisable to indicate not only the thickness of the resulting BDD film, but also its conductivity.
4. In section 2.1, you must indicate the brand of the silicon substrate and the brand of nanodiamonds. Describe how nanodiamonds were fixed on a silicon substrate.
5. The sentence “CoNiO2/Co2O3 arrays were conducted on BDD by CHI760E electrodeposition in a standard three-electrode system” (section 2.2) should be redone, since this is the first time the undeciphered combination of words “CHI760E electrodeposition” appears here. I suggest instead of “by CHI760E electrodeposition” write “using a CHI 760E model Electrochemical Workstation”.
6. In section 2.2, write explicitly at what stage the nitrates turned into oxides or hydroxides if annealing was carried out in an argon atmosphere.
7. If we are talking about CoNiO2/Co2O3 arrays, then it is obvious that the Co/Ni atomic ratio in these arrays will be greater than one. However, in section 2.2 it is written “Ni(NO3)2 6H2O (0.8 mmol), Co(NO3)2 6H2O (0.8 mmol) and NH4Cl (8 mmol) were mixed in 80 mL deionized water and transferred into 100 mL electrolytic cell". It is necessary to explain how the result is Co/Ni > 1. If the presence of Co2O3 is allowed, then the presence of NiOx must be allowed.
8. In section 2.4, you must indicate the brands of activated carbon, electrolyte, current collector, and separator. Indicate how and in what quantity the electrolyte was administered. It is advisable to present a design diagram of the SC with all actual dimensions.
9. Many will be interested to know what the fate of the silicon substrate is in the final SC design.
10. Lines 119-121: “Figures 1b-1d present the SEM images of CoNiO2/Co2O3/BDD composites with deposition time of 1000 s, 2500 s and 5000 s, respectively.” I would like to see similar SEM images after annealing at 300 C. They are of greater interest from the point of view of the design of the SC electrode. If Figure 1 shows SEM images of CoNiO2/Co2O3/BDD composites after annealing, then this should be written somewhere.
11. The inserts in Figure 1 are very small and not very clear. It would be desirable to bring them into SI on a larger scale. The SI could describe in more detail how the thicknesses of the CoNiO2/Co2O3 arrays were determined.
12. It is necessary to provide the accuracy of determining interplanar distances using TEM.
13. It would be good to calculate and report the Co/Ni ratio from EDS data obtained with a defocused electron beam or during scanning with a beam over a large area.
14. The text devoted to the analysis of XPS spectra is written extremely poorly. It is not clear what the “Gaussian fitting method” is (you need to provide a Ref where this method is described). The shape of the peaks that were used to describe the spectra is not indicated. It is unclear how the background was approximated. It is not clear why Figures 3b, 3c and 3d have different signal to noise ratios.
15. For an objective assessment of the qualitative composition of the surface, it is advisable to provide overview XPS spectra for both electrodes.
16. I do not agree with the conclusion (lines 173-174) “XPS results indicate the successful formation of CoNiO2/Co3O4 composites, which is complementary to XRD analysis.” From XPS it follows that Ni2+, Ni3+, Co3+, Co2+ ions are present on the surface of the electrode under study. Moreover, the presence of Ni3+ ions in the CoNiO2/Co2O3 structure is not formally assumed. Their education must be explained.
17. Line 230: “Figure 4 (a) CV curves at a scanning rate of 50 mV s−1”, and directly in Figure 5 it is written “5 mV s−1”.
18. In Figure 4f, the label on the Y-axis needs to be corrected.
19. I completely agree with the conclusion “Notably, the specific capacitance of CoNiO2/Co3O4/BDD exceeds most diamond-based supercapacitors.” This is not surprising, since the specific surface areas of diamond electrodes are very low. It is necessary to compare with the specific capacity of electrodes based on CoNiO2 or Co3O4.
20. Lines 226-228 “At a scan rate of 1.0 mV s−1, the capacitive contribution is 48%, and the diffusion-controlled process accounts for 52% (Figure S4).” This proposal contrasts with the conclusion about the speed of SC based on CoNiO2/Co3O4/BDD. Everyone knows that diffusion processes are slower than electronic ones.
21. Finally, I propose to soften the advertising optimism of the sentence “It is worth noting that a red light-emitting diode is lit directly by only one asymmetrical supercapacitor without any other power assistance, indicating that this electrode material has excellent application prospects.”
In conclusion, I present the plot of Ragone for energy storage devices from the article Journal of Energy Storage 73 (2023) 109293. Let the authors themselves put the values they obtained on this plot.

Author Response
Reviewer #3
A rather complex design of an asymmetric supercapacitor with rather modest values of specific energy and specific power is described. However, the results obtained appear to be reliable. Once the following issues have been addressed, the manuscript can be published.
The authors thank Reviewer for offering valuable suggestions and comments to improve the scientific level of manuscript. We have carefully considered these comments and suggestions and answered the questions accordingly.
Q1: In the introduction, it is necessary to explicitly indicate whether BDD films were previously used to create supercapacitors, and if they were, then indicate these works and the specific energy and power values obtained in them. This is necessary to understand both the degree of novelty and the level of results obtained in the work being reviewed.
Answer: Thanks for the reviewer’s kind suggestion. According reviewer’s suggestion, we have added previous reports for diamond-based supercapacitor electrodes and illustrated their power density and energy density. The related illustration is in lines 75-79 of page 2.
Q2: For an objective assessment of the work done, in section “2.1 Synthesis of boron doped diamond (BDD)” it is necessary to give the mass of hydrogen that must be consumed to obtain BDD film with a thickness of 20 μm.
Answer: Thanks for the reviewer’s kind suggestion. According to reviewer’s suggestion, the mass of hydrogen consumed to obtain a BDD film with a thickness of 20 μm is 25.63 g. The related illustration is in lines 110-111 of page 3.
Q3: It is also advisable to indicate not only the thickness of the resulting BDD film, but also its conductivity.
Answer: Thanks for the reviewer’s kind suggestion. According to the test of Hall effector, the conductivity of BDD is 113.63 S cm-1. The related illustration is in lines 111-112 of page 3.
Q4: In section 2.1, you must indicate the brand of the silicon substrate and the brand of nanodiamonds. Describe how nanodiamonds were fixed on a silicon substrate.
Answer: Thanks for your constructive comments. According to reviewer’s suggestion, silicon substrate and nano-diamond are purchased from Tianjin Jingchen Electronics Company and Tianjin Qianyu Superhard Technology Co., LTD, respectively. The related illustration is in lines 93-95 of page 2.
Q5: The sentence “CoNiO2/Co2O3 arrays were conducted on BDD by CHI760E electrodeposition in a standard three-electrode system” (section 2.2) should be redone, since this is the first time the undeciphered combination of words “CHI760E electrodeposition” appears here. I suggest instead of “by CHI760E electrodeposition” write “using a CHI 760E model Electrochemical Workstation”.
Answer: Thanks for reviewer’s kind comments. According reviewer’s suggestion, we have revised it to “CHI 760E model Electrochemical Workstation” in lines 115-116 of page 3.
Q6: In section 2.2, write explicitly at what stage the nitrates turned into oxides or hydroxides if annealing was carried out in an argon atmosphere.
Answer: Thanks for the reviewer’s kind suggestion. In the process of constant potential deposition, nitrate ion and hydroxide in solution were replaced to form nickel-cobalt hydroxide. The precursor of nickel-cobalt hydroxide was annealed in argon to form CoNiO2/Co2O3.
Q7: If we are talking about CoNiO2/Co2O3 arrays, then it is obvious that the Co/Ni atomic ratio in these arrays will be greater than one. However, in section 2.2 it is written “Ni(NO3)2 6H2O (0.8 mmol), Co(NO3)2 6H2O (0.8 mmol) and NH4Cl (8 mmol) were mixed in 80 mL deionized water and transferred into 100 mL electrolytic cell". It is necessary to explain how the result is Co/Ni > 1. If the presence of Co2O3 is allowed, then the presence of NiOx must be allowed.
Answer: Thanks for the reviewer’s kind suggestion. Since the electronegativity of cobalt is greater than that of nickel, the reaction is not completely in accordance with the feed ratio, and cobalt is easier to form the corresponding oxide than nickel, and nickel ions are easier to free in solution. Therefore, Co/Ni > 1 is obtained for the sample.
Q8: In section 2.4, you must indicate the brands of activated carbon, electrolyte, current collector, and separator. Indicate how and in what quantity the electrolyte was administered. It is advisable to present a design diagram of the SC with all actual dimensions.
Answer: Thanks for the reviewer’s kind suggestion. According to reviewer’s suggestion, we provide the brands of activated carbon, electrolyte, current collector, and separator in lines 142-145 of page 3. And we have added a design diagram of the SC with all actual dimensions in Figure S1.
Q9: Many will be interested to know what the fate of the silicon substrate is in the final SC design.
Answer: Thanks for the reviewer’s kind suggestion. The silicon substrate acts as a positive fluid collector in the device.
Q10: Lines 119-121: “Figures 1b-1d present the SEM images of CoNiO2/Co2O3/BDD composites with deposition time of 1000 s, 2500 s and 5000 s, respectively.” I would like to see similar SEM images after annealing at 300 C. They are of greater interest from the point of view of the design of the SC electrode. If Figure 1 shows SEM images of CoNiO2/Co2O3/BDD composites after annealing, then this should be written somewhere.
Answer: Thanks for the reviewer’s kind suggestion. Figures 1b-1d shows the SEM images of annealed CoNiO2/Co3O4/BDD composites with deposition time of 1000 s, 2500 s and 5000 s, respectively. We have revised the related illustration in lines 153-155 of page 4.
Q11: The inserts in Figure 1 are very small and not very clear. It would be desirable to bring them into SI on a larger scale. The SI could describe in more detail how the thicknesses of the CoNiO2/Co2O3 arrays were determined.
Answer: Thanks for the reviewer’s kind suggestion. According to reviewer’s suggestion, the inserts is included in the supporting information (Figure S2). The related descriptions in detail how to determine the thickness of CoNiO2/Co2O3 arrays are in lines 155-157 of page 4.
Q12: It is necessary to provide the accuracy of determining interplanar distances using TEM.
Answer: Thanks for the reviewer’s kind suggestion. According to reviewer’s suggestion, the layer spacing measured by TEM is 94 nm (Figure 2a). The related illustration is in lines 182-183 of page 4.
Q13: It would be good to calculate and report the Co/Ni ratio from EDS data obtained with a defocused electron beam or during scanning with a beam over a large area.
Answer: Thanks for the reviewer’s kind suggestion. Co/Ni ratio from EDS data obtained is shown in Table S2. The related illustration is in lines 188-189 of page 5.
Q14: The text devoted to the analysis of XPS spectra is written extremely poorly. It is not clear what the “Gaussian fitting method” is (you need to provide a Ref where this method is described). The shape of the peaks that were used to describe the spectra is not indicated. It is unclear how the background was approximated. It is not clear why Figures 3b, 3c and 3d have different signal to noise ratios.
Answer: Thanks for the reviewer’s kind suggestion. According to reviewer’s suggestion, the more detailed elemental composition and the oxidation state of the as-prepared CoNiO2/Co3O4 are further characterized by XPS measurements and the corresponding results are presented in Figure 3b–d. Binding energy is calibrated by fixing the saturated hydrocarbon component of the C1s peak at 284.8 eV. Gaussian function fitting is performed for all peaks. The related illustration is in lines 198-211 of page 5.
Q15: For an objective assessment of the qualitative composition of the surface, it is advisable to provide overview XPS spectra for both electrodes.
Answer: Thanks for the reviewer’s kind suggestion. According to reviewer’s suggestion, we provide XPS of BDD films in Figure S4. BDD has a small amount of sp2 carbon, which is conducive to improving the conductivity of the electrode. The related description is in lines 214-215 of page 5.
Q16: I do not agree with the conclusion (lines 173-174) “XPS results indicate the successful formation of CoNiO2/Co3O4 composites, which is complementary to XRD analysis.” From XPS it follows that Ni2+, Ni3+, Co3+, Co2+ ions are present on the surface of the electrode under study. Moreover, the presence of Ni3+ ions in the CoNiO2/Co2O3 structure is not formally assumed. Their education must be explained.
Answer: Thanks for the reviewer’s kind suggestion. According to reviewer’s suggestion, we have deleted this sentence. It's not exactly +3 or +2 because of the disproportionation that can happen during the reaction.
Q17: Line 230: “Figure 4 (a) CV curves at a scanning rate of 50 mV s−1”, and directly in Figure 5 it is written “5 mV s−1”.
Answer: Thanks for the reviewer’s kind suggestion. Sorry for our mistake, we have revised the description of the scan rate of CV in lines 221-222 of page 6.
Q18: In Figure 4f, the label on the Y-axis needs to be corrected.
Answer: Thanks for the reviewer’s kind suggestion. According reviewer’s suggestion, we have corrected the Y-axis in Figure 4f.
Q19: I completely agree with the conclusion “Notably, the specific capacitance of CoNiO2/Co3O4/BDD exceeds most diamond-based supercapacitors.” This is not surprising, since the specific surface areas of diamond electrodes are very low. It is necessary to compare with the specific capacity of electrodes based on CoNiO2 or Co3O4.
Answer: Thanks for the reviewer’s kind suggestion. According reviewer’s suggestion, we have made a comparison in Table S1. Cobalt-based compounds have a high capacity due to their high theoretical specific capacitance. However, due to the low potential window, the combination of diamond with a wider potential window will provide a good idea for improving the energy density.
Q20: Lines 226-228 “At a scan rate of 1.0 mV s−1, the capacitive contribution is 48%, and the diffusion-controlled process accounts for 52% (Figure S4).” This proposal contrasts with the conclusion about the speed of SC based on CoNiO2/Co3O4/BDD. Everyone knows that diffusion processes are slower than electronic ones.
Answer: Thanks for the reviewer’s kind suggestion. According reviewer’s suggestion, we added a conclusion comparing the SC velocity based on CoNiO2/Co3O4/BDD. As shown in Fig. S8b, with the scan rate increase, the capacitive contribution is even higher. It suggests that the capacitive contribution plays a dominant role in the total capacity, and a faraday redox reaction occurs mainly on the surfaces of CoNiO2/Co3O4/BDD nanostructures. The related description is in lines 271-274 of page 7.
Q21: Finally, I propose to soften the advertising optimism of the sentence “It is worth noting that a red light-emitting diode is lit directly by only one asymmetrical supercapacitor without any other power assistance, indicating that this electrode material has excellent application prospects.”
Answer: Thanks for the reviewer’s kind suggestion. According reviewer’s suggestion, we have revised this sentence to “It is worth noting that a red light-emitting diode is lit directly by only one asymmetrical supercapacitor without any other power assistance, indicating that this electrode material has excellent application prospects” in lines 305-307 of page 8.
Q22: In conclusion, I present the plot of Ragone for energy storage devices from the article Journal of Energy Storage 73 (2023) 109293. Let the authors themselves put the values they obtained on this plot.
Answer: Thanks for reviewer’s kind comments. According reviewer’s suggestion, we have compared it with other literature in Figure S10. The relevant description is in lines 299-305 of page 8. The relevant literature is cited and numbered as 41.

Round 2
Reviewer 1 Report
Comments and Suggestions for Authors
The revised version is suitable for publication.
Reviewer 2 Report
Comments and Suggestions for Authors
The authors have adequately addressed all the queries raised. The revised manuscript can be accepted in its current form.
Comments on the Quality of English LanguageThe quality of English is good.
Reviewer 3 Report
Comments and Suggestions for Authors
Review No. 2
The authors agreed in principle with my remarks. If the editors agree with their answers and corrections, then I will also not object to the publication of the manuscript.